# Breakdown of energy transfer gap laws revealed by full-dimensional quantum scattering between HF molecules

Dongzheng Yang [1], Jing Huang [1], Xixi Hu [1]*, Hua Guo[2] & Daiqian Xie [1]*

Inelastic collisions involving molecular species are key to energy transfer in gaseous environments. They are commonly governed by an energy gap law, which dictates that transitions are dominated by those between initial and final states with roughly the same ro-vibrational energy. Transitions involving rotational inelasticity are often further constrained by the rotational angular momentum. Here, we demonstrate using full-dimensional quantum scattering on an *ab initio* based global potential energy surface (PES) that HF–HF inelastic collisions do not obey the energy and angular momentum gap laws. Detailed analyses attribute the failure of gap laws to the exceedingly strong intermolecular interaction. On the other hand, vibrational state-resolved rate coefficients are in good agreement with existing experimental results, validating the accuracy of the PES. These new and surprising results are expected to extend our understanding of energy transfer and provide a quantitative basis for numerical simulations of hydrogen fluoride chemical lasers.

[1] Institute of Theoretical and Computational Chemistry, Key Laboratory of Mesoscopic Chemistry, School of Chemistry and Chemical Engineering, Nanjing University, Nanjing 210023, China. [2] Department of Chemistry and Chemical Biology, University of New Mexico, Albuquerque, New Mexico 87131, USA. *email: xxhu@nju.edu.cn; dqxie@nju.edu.cn

Collision-induced energy transfer is a fundamentally important process in many gas-phase chemical environments, such as combustion[1], atmospheric chemistry[2], astrochemistry[3], and chemical laser engineering[4]. It is responsible for a wide array of macroscopic phenomena, such as heating and cooling as well as energy transport. As a result, many experimental and theoretical studies have been devoted to a better understanding of such processes[5,6]. Since molecules are quantum objects, the ultimate understanding need be based on inelastic transitions between different quantum states of the collision partners, which can now be prepared experimentally with lasers[6]. This field is currently undergoing a renaissance, thanks to advances in experimental techniques[7–12].

Many quantum state-resolved studies have found that the most dominant ro-vibrationally inelastic transitions are often resonant or nearly resonant[13,14]. A classic example is the resonant energy transfer between $CO_2(v_3 = 0, 1)$ and $N_2(v = 1, 0)$ in the $CO_2$ laser[15]. The resonant energy transfer is even more prevalent for highly excited molecules because of their higher densities of states[16]. These observations have led to rules of thumb on the dominant energy transfer mechanisms[6,17], often called gap laws[18,19]. The most important one is perhaps the energy gap law, which states that the most dominant inelastic transitions are the ones with the smallest energy gaps between the combined initial and final states of the collision partners. For many inelastic collisions involving rotational transitions, the rotational angular momentum presents an additional constraint[20], and these transitions might also be subjected to the angular momentum gap law, namely, the dominant transitions being the ones with the minimal changes in the sum of rotational angular momenta. Numerous experimental and theoretical studies to date have shown that molecular collisions conform to these propensity rules. As a result, most kinetic simulations are based on these gap laws to describe energy transfer when experimental or theoretical data are lacking.

As a working medium for chemical lasers[21], vibrational relaxation rates of hydrogen fluoride (HF) are of great interest to laser engineering, because they directly determine populations of HF ro-vibrational levels in the laser cavity[22]. The rate coefficient for self-relaxation of HF from its first excited vibrational state,

$$HF(v_1 = 1) + HF(v_2 = 0) \rightarrow HF(v_1' = 0) + HF(v_2' = 0). \quad (1)$$

has been extensively studied experimentally[23–29]. Theoretical understanding has unfortunately lagged behind. So far, dynamical calculations for HF self-relaxation have only been performed in 1970s[30–33], employing the quasi-classical trajectory (QCT) approximation on an analytical potential energy surface (PES) fit to 294 low-level self-consistent-field ab initio points[34]. The first quantum mechanical calculations[35] for vibrational energy transfer were carried out in 1983, with a rotation-neglected three-dimensional (3D) approximation. These theoretical studies are not expected to be quantitatively accurate because of the poor quality PES and inaccurate dynamical treatments.

So far, a full six-dimensional (6D) quantum dynamics study of the HF vibrational self-relaxation is still absent, despite the availability of full-dimensional PESs[36–38]. The slow progress can be largely attributed to difficulties associated with time-independent quantum mechanical (TIQM) calculations of ro-vibrationally inelastic scattering between two "heavy" molecules. Such calculations require much higher computational costs than those for systems involving light molecules such as $H_2$, because of (i) a large number of rotational levels in the basis set due to the small rotational constants of the "heavy" molecules, (ii) numerous partial waves contributing to the cross-section and the rate coefficient, and (iii) many propagation steps in solving the scattering equation. In the current system, such calculations are

particularly challenging because of the exceptionally strong intermolecular interaction, which makes it very difficult for a fully coupled quantum-scattering method to converge the cross-section, even when reaction probabilities at low $J$ partial waves are possible. To solve this problem, we have recently developed an approximate method, in which the coupled-states approximation (CSA) is improved with the nearest neighbor Coriolis couplings (NNCC)[39]. This CSA–NNCC method has been shown to significantly reduce computational costs without a major loss of accuracy. Its application to the HF–HF system reported here represents the first full-dimensional TIQM calculations of ro-vibrational inelastic scattering between two non-hydrogen molecules that yield converged cross-sections and rate coefficients, which can be directly compared with the experiment.

In this work, we present an accurate theoretical dynamics investigation on vibrationally inelastic scattering between two HF molecules. Our full-dimensional quantum-scattering calculations represent the first such study involving two "heavy" (non-$H_2$) molecules, and were made possible by a recently proposed scheme to reduce the size of the scattering calculations[39]. Both state-to-state cross-sections and rate coefficients are obtained, and the vibrational state-resolved rate coefficients are in excellent agreement with available experimental results, thanks to a highly accurate global PES obtained recently from high-level ab initio calculations[40]. Surprisingly, our results indicate that the energy transfer between these molecules satisfies neither the energy gap law nor angular momentum gap law. As discussed below, such breakdown might have significant implications in HF-based chemical lasers. We further demonstrate that the breakdown of these gap laws stems from the strong interaction between two HF molecules due to hydrogen bonding. This study thus expands our understanding of quantum state-resolved energy transfer between molecules.

## Results

**State-to-state and total rate coefficients.** For convenience, a notation labeled by four quantum numbers $(v_1, j_1; v_2, j_2)$ is used to describe a combined molecular state (CMS), which is the combination of ro-vibrational states of two diatoms before or after a collision[18]. Note that this CMS notation obeys the "well-ordered states" classification[41], in which $v_1 \geq v_2$ and $j_1 \geq j_2$ for $v_1 = v_2$. Similarly, the notation $(v_1; v_2)$ is used to label a combined vibrational state (CVS).

The state-to-state rate coefficient for the HF–HF collision from the initial $(1, 0; 0, 0)$ CMS was calculated as a function of temperature for each relevant final CMS. As shown in Fig. 1a for the rotationally elastic channel $j_2 = j_2' = 0$, the rate coefficients initially increase with temperature from 100 to 200 K, before decreasing from 200 to 1200 K. The only exception is for the final $(0, 14; 0, 0)$ CMS, whose rate coefficient increases monotonically. The rate coefficient to the final $(0, 13; 0, 0)$ CMS dominates in all the temperatures considered, followed by those to the $(0, 12; 0, 0)$ and $(0, 10; 0, 0)$ CMSs. The rate coefficient of the rotational–angular–momentum–conserving final CMS $(0, 0; 0, 0)$ is clearly suppressed, two orders of magnitude smaller than the largest one. Figure 1b shows the rate coefficients for the initial $(1, 3; 0, 0)$ CMS. Almost all the rate coefficients decrease monotonously with temperature. They increase with increasing $j_1'$, and the transition to $(0, 14; 0, 0)$ is dominant in this group.

In general, these state-to-state rate coefficients appear to be quite sensitive to the final rotational quantum numbers, but not to the temperature. The differences between rate coefficients of a single transition at 100 K and that at 1200 K are smaller than an order of magnitude, while the difference between that to the final $(0, 1; 0, 0)$ CMS and that to the final $(0, 14; 0, 0)$ can reach up to a factor of

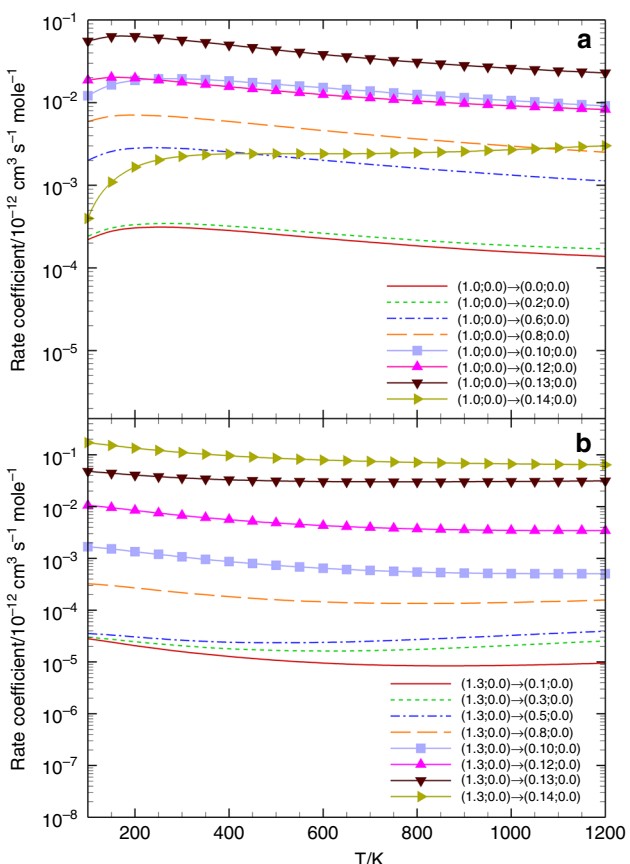

**Fig. 1** State-to-state rate coefficient as a function of temperature. Calculated results by applying Eq. (4) are presented for transitions $(1, 0; 0, 0) \rightarrow (0, j_1'; 0, 0)$ in panel **a** and $(1, 3; 0, 0) \rightarrow (0, j_1'; 0, 0)$ in panel **b**, respectively

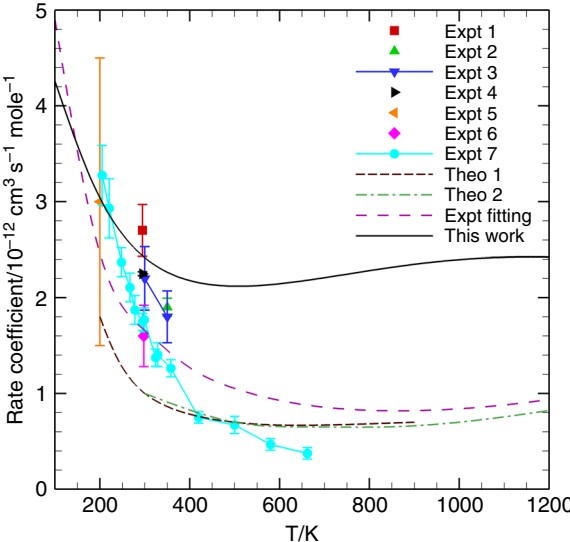

**Fig. 2** Calculated vibrational-resolved $(1; 0) \rightarrow (0; 0)$ rate coefficient as a function of temperature. The current calculated results are compared with available experimental (denoted as Expt 1[23], Expt 2[24], Expt 3[25], Expt 4[26], Expt 5[27], Expt 6[28], Expt 7[29], and Expt fitting[22]) and other theoretical results (Theo 1[31] and Theo 2[32]). All error bars represent the experimental error

$10^4$. Unfortunately, there are neither experimental nor theoretical data on such state-to-state rate coefficients to compare with.

**Vibrational state-resolved rate coefficients**. The vibrational state-resolved $(1; 0) \rightarrow (0; 0)$ rate coefficient was calculated by Boltzmann averaging the state-to-state rate coefficients from 256 initial CMSs $(1, j_1 = 0 \sim 15; 0, j_2 = 0 \sim 15)$. The result is shown in Fig. 2, together with the available experimental[23–29] and previous theoretical[31,32] data. Our quantum mechanical rate coefficient initially decreases with temperature from 100 to 500 K, but increases slowly with temperature from 500 to 1200 K, with a minimum value of $2.1 \times 10^{-12}$ cm$^3$ s$^{-1}$ mole$^{-1}$ at 500 K. The calculated rate coefficient $(3.03 \times 10^{-12}$ cm$^3$ s$^{-1}$ mole$^{-1}$) is in quite good agreement with the experimental result of $3.0 \times 10^{-12}$ cm$^3$ s$^{-1}$ mole$^{-1}$ at 200 K, reported by Hancock and Green[23]. At the room temperature (298 K), our result of $2.4 \times 10^{-12}$ cm$^3$ s$^{-1}$ mole$^{-1}$ is also consistent with the experiment by Ahl and Cool ($2.2 \times 10^{-12}$ cm$^3$ s$^{-1}$ mole$^{-1}$)[25] and by Bott ($2.25 \times 10^{-12}$ cm$^3$ s$^{-1}$ mole$^{-1}$)[26]. The overall agreement with experiment in the high temperature range is quite satisfactory, especially when the large variation of experimental data is taken into consideration. Our quantum rate coefficient indicates that all previous theoretical results, which were obtained using a semiclassical trajectory method on a much less accurate PES[31,32], underestimate at all temperatures. It is not clear what are the sources of the errors, but it is known that these theoretical results were obtained using less accurate PESs and approximate dynamical methods.

It is particularly interesting to compare our results with the rate data extracted from the chemical laser kinetic package[22], which

were obtained by fitting various experimental results. At low temperatures, the fitted results are consistent with our calculated ones, but they gradually deviate from ours as temperature increases, with a factor of 2.4 at 1200 K. Since our calculations are based on highly accurate electronic structure data and a full-dimensional quantum mechanical treatment of the scattering, we are confident about the accuracy of our calculated results in providing important benchmarks for chemical laser engineering.

**Breakdown of gap laws**. It is well established that collisional energy transfer between molecules is often governed by the energy and angular momentum gaps between the initial and final states[17]. In short, transitions are dominated by those with small gaps in the internal energy and/or in the rotational angular momentum, while others are suppressed[17]. Furthermore, mass-symmetric species (e.g., $H_2$) usually follow the angular momentum gap law, while those mass-asymmetric molecules like hydrogen halides tend to follow the energy gap law[17]. The conformation to these gap laws has also been extensively discussed and verified in the $H_2$–$H_2$[18,19,42], $H_2$–CO[43,44], $H_2$–CN[45], and Ar–HCl[46] systems. The cross-sections or rate coefficients of other HF-involved vibrational relaxation systems (Ar–HF and $H_2$–HF)[47,48] are also dominated by specific final CMSs whose energies nearly equal to the initial ones, in accord with the energy gap law.

In order to test the gap laws discussed above, state-to-state cross-sections for the initial $(1, 0; 0, 0)$ CMS are calculated, for which there are 83, 91, and 97 open final CMSs in total at the collision energy of 0.01, 0.05, and 0.1 eV, respectively. In Fig. 3, the cross-sections of these final CMSs are arranged by their energy differences with the initial CMS, from low to high. In all collision energies, it is clear that there is no final CMS dominating the cross-sections. Instead, inelastic transitions are spread out to a range of final CMSs with energy differences roughly between $-0.2$ and $-0.1$ eV. There is clearly a breakdown of the energy gap law.

The product rotational branching ratio for the initial $(1, 0; 0, 0)$ CMS is further examined in Fig. 4, which is evaluated in terms of

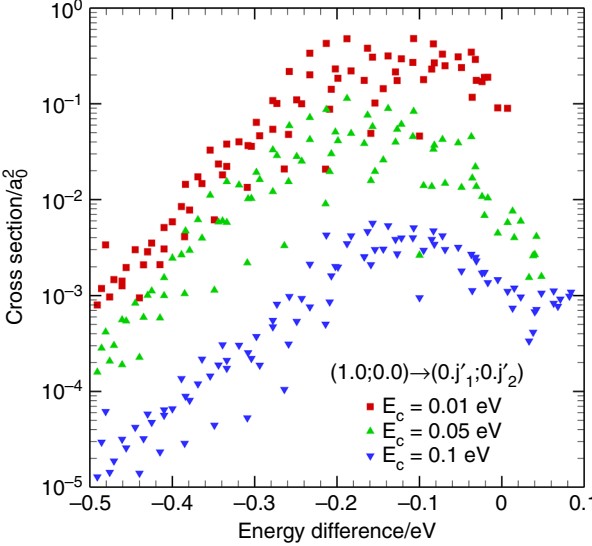

**Fig. 3** State-to-state cross sections for transitions $(1, 0; 0, 0) \rightarrow (0, j'_1; 0, j'_2)$. The results of all the corresponding open channel of final CMSs for vibrational relaxation are presented, showing as a function of the energy difference between final and initial CMSs at collision energies of 0.01, 0.05, and 0.1 eV, respectively

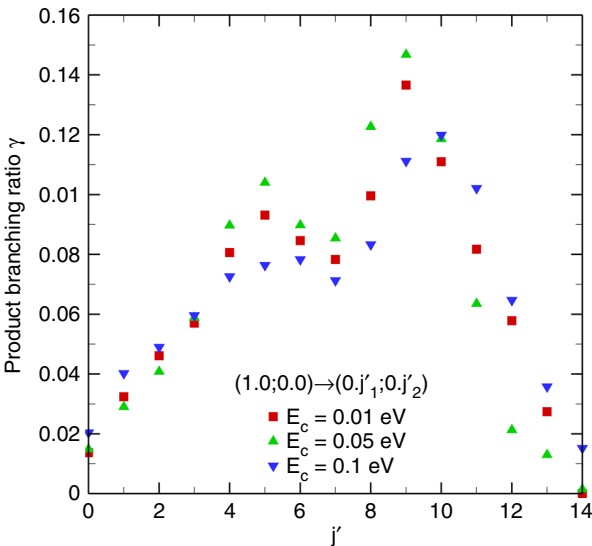

**Fig. 4** The product rotational branching ratio for the initial $(1, 0; 0, 0)$ CMS. All the products in the current energy transfer process reach the final state $(v' = 0, j')$. See Eq. (2) for details

the cross-section,

$$\gamma_{j'}^{v_1 j_1 v_2 j_2} = \frac{\sum_{j'_1=j'} \ \sum_{j'_2=j'} \left(1 + \delta_{j'_1 j'_2}\right) \sigma_{v'_1 j'_1 v'_2 j'_2 \leftarrow v_1 j_1 v_2 j_2}}{\sum_{j'_1 j'_2} \left(1 + \delta_{j'_1 j'_2}\right) \sigma_{v'_1 j'_1 v'_2 j'_2 \leftarrow v_1 j_1 v_2 j_2}}. \quad (2)$$

At collision energies of 0.01 and 0.05 eV, the most probable rotational quantum numbers of the final states tend to be $j' = 8 \sim 10$. As the collision energy increases, higher rotational channels are opened and populated. The most probable final states are $j'_{HF} = 9 \sim 10$ at the collision energy of 0.1 eV. Apparently, these data do not support the angular momentum gap law.

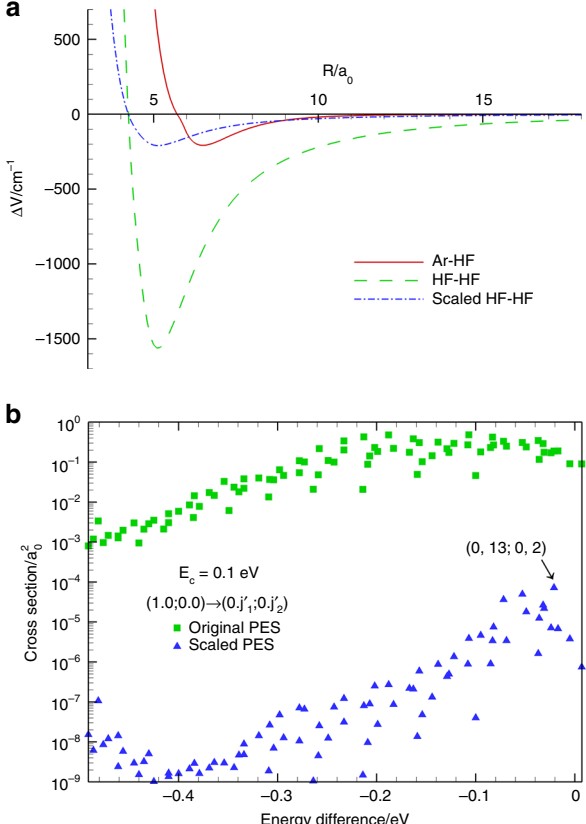

**Fig. 5** The results of using scaled and original potentials. **a** Comparison of three interaction potential energy curves as a function of the intermolecular distance, in which the intramolecular bond lengths are fixed to their equilibrium values. The HF–HF artificial PES is scaled from the original one by a factor of 0.134, which matches the depth of the Ar–HF potential. **b** Comparison of state-to-state cross-sections for transitions $(1, 0; 0, 0) \rightarrow (0, j'_1; 0, j'_2)$ at $E_c = 0.01$ eV, using the original interaction PES and the scaled one

Although the angular momentum gap law was not observed in HF vibrational relaxation by other colliders such as Ar[47] and H$_2$[48], these scattering processes were found to follow the energy gap law. The energy gap law in these systems guarantees that the energy of HF drains slowly away from vibration to translation, and this property has been exploited in the chemical lasers[17]. For the HF–HF system, however, rotational excitations of the products are somewhat "colder" than other systems, which might have significant implications in chemical laser engineering.

We hypothesize that the surprising rule-breaking behaviors stem from the extraordinarily strong intermolecular interaction, which form a strong hydrogen bond. To this end, we carried out the quantum dynamics calculations using an artificially PES scaled from the ab initio HF–HF PES by a factor of 0.134, which makes the depth of the interaction potential equal to that of the Ar–HF system, as shown in Fig. 5a. Figure 5b compares the cross-section for the initial $(1, 0; 0, 0)$ CMS at $E_c = 0.01$ eV, using the original PES and the scaled one. The cross-sections on the scaled PES are overall three to four orders of magnitude smaller than those on the unscaled one, due apparently to a shallow interaction well and weaker inter-vibrational couplings. Interestingly, the results on the scaled PES show that the three final CMSs close to zero energy difference dominate. The most dominant final CMS is $(0, 13; 0, 2)$, marked in the figure, which can be likened to the

largest cross-section of final state ($v' = 0$, $j' = 13$) of HF in the Ar–HF system, as shown in previous studies[46,47]. This result strongly supports our hypothesis that the breakdown of the energy gap law in the current system is indeed due to the deep interaction well.

It is conceivable that the attractive part of the PES provides ample opportunity for energy randomization, thanks to the high density of states in the well. To illustrate this point, we have carried out QCT calculations for HF–HF scattering at low collision energies. As shown in Supplementary Fig. 3, a significant fraction of the scattering trajectories has multiple inner turning points and long lifetimes, suggesting significant trapping. This is consistent with the previous QCT results for predissociation of the (HF)$_2$ dimer[49]. Such energy scrambling is not possible for scattering on PESs with shallow wells. Under the latter circumstances, the energy transfer is dominated by impulsive collisions due to repulsive walls in short distances, on which the gap laws are based[17].

## Discussion

In this work, we report the converged full-dimensional quantum mechanical cross-sections for the HF–HF inelastic scattering on a newly developed global PES based on high-level ab initio calculations. The calculated vibrational energy transfer rate coefficient is in reasonably good agreement with existing experimental results, validating the PES. These quantum state-resolved rate coefficients provide useful information for chemical laser engineering. Furthermore, our calculations found that the energy and angular momentum gap laws are not observed in this system. The breakdown of the well-established propensity rules is attributed to the extraordinarily strong intermolecular interaction between the two dipolar molecules, which leads to energy transfer in the attractive potential well rather than on the repulsive potential wall. The failure of these well-established propensity rules suggests a different regime in energy transfer, which might need be carefully considered in kinetic simulations involving other strongly interacting systems, such as dimers formed by hydrogen bonding molecules (e.g., $H_2O$ and $NH_3$).

## Methods

**Quantum dynamics calculation.** The details of the TIQM scattering calculations are given in our previous work[39] and in Supplementary Information (SI) as well, thus only a brief outline is given here. The Hamiltonian is defined in diatom–diatom Jacobi coordinates and the scattering matrix (S-matrix) elements for a given set of total angular momentum ($J$) and parities are computed in the body-fixed frame using a log-derivative method[50,51]. The validity of CSA–NNCC approach has been demonstrated for the H$_2$–H$_2$/HD[39], and H$_2$–HF[39,52] systems, and in SI for the current system as well. The use of this approximation reduces the size of the system to $J_{max}/3$, which makes such calculations possible. Since TIQM scales as $N^3$, where $N$ is the dimension of the Hamiltonian matrix, the saving amounts to about $10^3$ for $J = 150$. Our recently developed 6D PES[40] is used, which is much more accurate for scattering studies than previous PESs because it is based on high-level ab initio data, free of empirical adjustments, and accurate in both strongly interacting and in asymptote regions.

The state-to-state integral cross-section can be evaluated in terms of the state-to-state probability $P^J$,

$$\sigma_{v'_1 j'_1 v'_2 j'_2 \leftarrow v_1 j_1 v_2 j_2}(E_c) = \frac{\pi}{k^2_{v_1 j_1 v_2 j_2}} \sum_J (2J+1) P^J_{v'_1 j'_1 v'_2 j'_2 \leftarrow v_1 j_1 v_2 j_2}(E_c), \quad (3)$$

where $k_{v_1 j_1 v_2 j_2} = \sqrt{2\mu E_c}$ is the translational wave vector, and $E_c$ is the collision energy. The rate coefficient as a function of temperature $T$ is given by

$$k_{v'_1 j'_1 v'_2 j'_2 \leftarrow v_1 j_1 v_2 j_2}(T) = \frac{1}{k_B T} \sqrt{\frac{8}{\pi \mu k_B T}} \int_0^\infty \sigma_{v'_1 j'_1 v'_2 j'_2 \leftarrow v_1 j_1 v_2 j_2}(E_c) \exp(-E_c/k_B T) E_c dE_c, \quad (4)$$

where $k_B$ is the Boltzmann constant and $\mu$ is the reduced mass of this system. The total vibrational relaxation rate coefficient for an initial CMS can be obtained by simply summing the state-to-state rate coefficients over all the corresponding final rotational states,

$$k_{v'_1 v'_2 \leftarrow v_1 j_1 v_2 j_2}(T) = \sum_{j'_1 j'_2} k_{v'_1 j'_1 v'_2 j'_2 \leftarrow v_1 j_1 v_2 j_2}(T). \quad (5)$$

In order to compare with experimental results, which do not distinguish rotational states, the vibrational state-resolved rate coefficient is calculated by Boltzmann averaging the $k_{v'_1 v'_2 \leftarrow v_1 j_1 v_2 j_2}$ over all the initial rotational states,

$$k_{v'_1 v'_2 \leftarrow v_1 v_2}(T) = \frac{\sum_{j_1 j_2} w_{v_1 j_1} w_{v_2 j_2} k_{v'_1 v'_2 \leftarrow v_1 j_1 v_2 j_2}(T)}{\sum_{j_1 j_2} w_{v_1 j_1} w_{v_2 j_2}}, \quad (6)$$

where the weight factors $w_{v_j i} = (2j_i + 1)\exp\left(-E_{v_j i}/k_B T\right)$ and $E_{vj}$ are the ro-vibrational energy.

**Parameters for convergence.** Numerous convergence tests have been performed, and the final parameters used in the calculations are given here. Specifically, all propagations were carried out with a variable radial step size $\Delta R$ in different ranges of $R$, namely $\Delta R = 0.02$ $a_0$ in $R \in [3.5, 6.0]$ $a_0$, $0.03$ $a_0$ in $R \in [6.0, 20.0]$ $a_0$, and $0.05$ $a_0$ in $R \in [20.0, 45.0]$ $a_0$, respectively, where $a_0$ denotes as the Bohr radius. A sufficient large distance of $R = 80.0$ $a_0$ was applied in the test calculations and the results showed that the state-to-state probabilities change no >1%. The number of potential optimized discrete variable representation (PODVR)[53] points were chosen as $N_{r_1} = N_{r_2} = 4$. The number of points in $\theta_1$ and $\theta_2$ for Gauss–Legendre quadrature and in $\phi$ for Chebyshev quadrature were chosen as $N_{\theta_1} = 30$, $N_{\theta_2} = 30$, and $N_\phi = 15$, respectively.

Three initial CVSs were considered, namely (0; 0), (1; 0), and (1; 1). Different maximum values of the sum of rotational quantum numbers $(j_1 + j_2)_{max}$ were introduced in the calculations to limit the number of the internal states included in the basis set, namely $(j_1 + j_2)_{max} = 35$, 20, and 10, respectively, for the three initial CVSs. These choices of the basis set finally made the number of coupled channels in the typical calculations to be 1450, 5925, and 12685 for $J(P_{ex}p\varepsilon) = 0(+++)$, $5(+++)$, and $10(+++)$, respectively.

## Data availability

The data that support the findings of this study are available from the corresponding author upon reasonable request.

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

## Acknowledgements
This work is supported by the National Key Research and Development Program of China (2017YFA0206500 to D.X.), the National Natural Science Foundation of China (Grant nos. 21590802 and 21733006 to D.X., U1932147 to X.H.), and the US Army Research Office (Grant No. W911NF-19-1-0283 to H.G.). All the dynamical calculations have been done on the computing facilities in the High-Performance Computing Center (HPCC) of Nanjing University. H.G. thanks Bala Naduvalath for some useful discussion.

## Author contributions
D.Y., X.H., D.X., and H.G. conceived the research. D.Y. performed the quantum dynamics calculations. J.H. constructed the potential energy surface. X.H. performed the QCT analysis. All authors contributed to the discussion and the writing, and approved the paper.

## Competing interests
The authors declare no competing interests.
