## [Peer Review File · Nature Communications]

**Breakdown of Energy Transfer Gap Laws Revealed by Full-Dimensional Quantum
Scattering between HF Molecules**

Yang et al

Supplementary Fig. 1 Body-fixed Jacobi coordinates ($R, r_1, r_2, \theta_1, \theta_2, \phi$) for the HF-HF system.

Supplementary Fig. 2 RMSEs of the state-to-state probabilities from various initial CMSs. Only probabilities obtained by using rigorous CC approach that are larger than 0.01 are included. The total energy is fixed at $E=1.133$ eV and the total angular momentum and the parity are chosen as $J(P_{\text{exp}}\epsilon)=10(+--+)$. There are 107 initial CMSs at this total energy and they are arranged in the sequence of the sum of two diatomic rotational quantum numbers j_1+j_2 from a high value to a low one.

Supplementary Fig. 3 The lifetime distributions for trajectories with different numbers of inner turning points. The last column include all trapped trajectories at the end of the 10 ps propagation.

Supplementary Notes 1

For diatom–diatom scattering problems, it is convenient to use the diatom-diatom Jacobi coordinate system $(R, r_1, r_2, \theta_1, \theta_2, \phi)$, as shown in Supplementary Figure 1. The Hamiltonian in this coordinate system has the following form,

$$H = -\frac{\hbar^2}{2\mu} \frac{\partial^2}{\partial R^2} + \frac{(\mathbf{J} - \mathbf{j}_{12})^2}{2\mu R^2} + h_1(r_1) + h_2(r_2) + \Delta V(R, r_1, r_2, \theta_1, \theta_2, \phi), \quad (1)$$

where μ is the reduced mass of the HF–HF system and ΔV is the interaction potential between two diatoms. $h_i(r_i)$ are diatomic ro-vibrational Hamiltonians expressed as

$$h_i(r_i) = -\frac{\hbar^2}{2\mu_i} \frac{\partial^2}{\partial r_i^2} + \frac{\mathbf{j}_i^2}{2\mu_i r_i^2} + V_i(r_i), \quad (2)$$

where μ_i are the reduced masses of the diatoms. The corresponding Schrödinger equations are $h_i |v_i j_i\rangle = E_{v_i j_i} |v_i j_i\rangle$ ($i = 1, 2$), in which $|v_i j_i\rangle$ are the ro-vibrational eigenfunctions labeled by the vibrational and rotational quantum numbers v_i and j_i , respectively, and $E_{v_i j_i}$ are their energy eigenvalues.

The total scattering wave function can be expanded as

$$|\Psi^{\pm J p \varepsilon}\rangle = \sum_{vjK} F_{vjK}^{\pm J p \varepsilon}(R) |X_{vjK}^{\pm J p \varepsilon}\rangle, \quad (3)$$

where the symmetrized body-fixed (BF) ro-vibrational functions serve as the basis for the system including two identical molecules,¹

$$|X_{vjK}^{\pm J p \varepsilon}\rangle = \Delta_{vj, \tilde{v}\tilde{j}} \left[|Y_{jK}^{J p \varepsilon}\rangle |\Phi_{vj}\rangle \pm \varepsilon (-1)^{j_{12}} |Y_{\tilde{j}K}^{J p \varepsilon}\rangle |\Phi_{\tilde{v}\tilde{j}}\rangle \right]. \quad (4)$$

The normalization factor $\Delta_{vj, \tilde{v}\tilde{j}} = \left[2(1 + \delta_{v\tilde{v}} \delta_{j\tilde{j}}) \right]^{-1/2} = \left[2(1 + \delta_{v_1 v_2} \delta_{j_1 j_2}) \right]^{-1/2}$. The radial basis is a direct product of two diatomic ro-vibrational eigenfunctions,

$$|\Phi_{v_j}\rangle = |v_1 j_1\rangle |v_2 j_2\rangle, \quad (5)$$

and $|Y_{jK}^{Jp\varepsilon}\rangle$ are parity-adapted BF total angular momentum eigenfunctions in the coupled angular momentum representation²

$$|Y_{jK}^{Jp\varepsilon}\rangle = \frac{1}{\sqrt{2 + 2\delta_{K0}}} \left[|JMK\rangle |j_1 j_2 j_{12} K\rangle + \varepsilon (-1)^{J+j_1+j_2+j_{12}} |JM(-K)\rangle |j_1 j_2 j_{12}(-K)\rangle \right]. \quad (6)$$

In Supplementary equation (6), $|JMK\rangle$ is the normalized Wigner rotation matrix defined in terms of the Euler angles³

$$|JMK\rangle = \sqrt{\frac{2J+1}{8\pi^2}} D_{K,M}^{J*}(\alpha, \beta, \gamma), \quad (7)$$

where M and K are the projections of total angular momentum J onto the space-fixed (SF) and BF z axis, respectively⁴. And $|j_1 j_2 j_{12} K\rangle$ is expressed as

$$|j_1 j_2 j_{12} K\rangle = \sum_m \langle j_1 m j_2 (K-m) | j_{12} K \rangle Y_{j_1}^m(\theta_1, 0) Y_{j_2}^{K-m}(\theta_2, \phi), \quad (8)$$

where $\langle \dots | \dots \rangle$ is the Clebsch-Gordan coefficient and $Y_j^m(\theta, \phi)$ is the spherical harmonic function³. In this representation, \mathbf{j}_1 and \mathbf{j}_2 are coupled to \mathbf{j}_{12} , which is subsequently coupled with the orbital angular momentum \mathbf{l} to yield the total angular momentum \mathbf{J} . For the total parity $p=+1$, the quantum number K ranges from 0 to J ; otherwise ($p=-1$), K ranges from 1 to J . Note the restriction that $p=+1$ for $J=0$, and that for $v_1=v_2$ and $j_1=j_2$ case, the allowed j_{12} quantum numbers must satisfy the condition $P_{\text{ex}} \varepsilon (-1)^{j_{12}} = 1$ ⁵. We used the collective indices $v \equiv (v_1, v_2)$ and $j \equiv (j_1, j_2, j_{12})$, while \tilde{v} and \tilde{j} denote exchange of the corresponding indices respectively, i.e. $\tilde{v} \equiv (v_2, v_1)$ and $\tilde{j} \equiv (j_2, j_1, j_{12})$.

Substituting the Hamiltonian and the basis functions into the time-independent

Schrödinger equation leads to the close-coupling (CC) equations,

$$\left(\frac{d^2}{dR^2} + k_{v_1 j_1 v_2 j_2}^2 \right) F_{v j K}^{\pm J p \varepsilon} = \sum_{v' j' K'} \left(2\mu \mathbf{V}_{v j K, v' j' K'}^{\pm J p \varepsilon} + \frac{1}{R^2} \mathbf{U}_{v j K, v' j' K'}^{\pm J p \varepsilon} \right) F_{v' j' K'}^{\pm J p \varepsilon}, \quad (9)$$

in which $k_{v_1 j_1 v_2 j_2} = \sqrt{2\mu(E - E_{v_1 j_1} - E_{v_2 j_2})}$ is the channel wave number and E is the total energy of

this system. The symmetrized interaction potential matrix $\mathbf{V}^{\pm J p \varepsilon}$ and the centrifugal matrix $\mathbf{U}^{\pm J p \varepsilon}$

can be written explicitly in terms of their corresponding unsymmetrized counterparts, respectively,

$$\mathbf{V}_{v j K, v' j' K'}^{\pm J p \varepsilon} = 2\Delta_{v_1 j_1, v_2 j_2} \Delta_{v'_1 j'_1, v'_2 j'_2} \left[\mathbf{V}_{v j K, v' j' K'}^{J p \varepsilon} \pm \varepsilon (-1)^{j_{12}} \mathbf{V}_{\tilde{v} j K, v' j' K'}^{J p \varepsilon} \right], \quad (10)$$

$$\mathbf{U}_{v j K, v' j' K'}^{\pm J p \varepsilon} = 2\Delta_{v_1 j_1, v_2 j_2} \Delta_{v'_1 j'_1, v'_2 j'_2} \left[\mathbf{U}_{v j K, v' j' K'}^{J p \varepsilon} \pm \varepsilon (-1)^{j_{12}} \mathbf{U}_{\tilde{v} j K, v' j' K'}^{J p \varepsilon} \right], \quad (11)$$

where the unsymmetrized one can be calculated by⁵

$$\mathbf{V}_{v j K, v' j' K'}^{J p \varepsilon} = \delta_{KK'} \langle v_1 j_1 | \langle v_2 j_2 | \langle j_1 j_2 j_{12} K | \Delta V | j'_1 j'_2 j'_{12} K' \rangle | v'_2 j'_2 \rangle | v'_1 j'_1 \rangle, \quad (12)$$

$$\begin{aligned} \mathbf{U}_{v j K, v' j' K'}^{J p \varepsilon} &= \delta_{vv'} \delta_{jj'} \left\{ \delta_{KK'} \left[J(J+1) + j_{12}(j_{12}+1) - 2K^2 \right] \right. \\ &\quad \left. - \delta_{K', K+1} \sqrt{1 + \delta_{K,0}} \lambda_{JK}^+ \lambda_{j_{12}K}^+ - \delta_{K', K-1} \sqrt{1 + \delta_{K,1}} \lambda_{JK}^- \lambda_{j_{12}K}^- \right\}, \end{aligned} \quad (13)$$

in which $\lambda_{JK}^{\pm} = \sqrt{J(J+1) - K(K \pm 1)}$ and $\lambda_{j_{12}K}^{\pm} = \sqrt{j_{12}(j_{12}+1) - K(K \pm 1)}$. The CC equations

can be solved effectively by applying the log-derivative method developed by Johnson⁶ and

Manolopoulos⁷, with a set of given $J, P_{\text{ex}}, p, \varepsilon$ and total energy E . In this method, the log derivative

matrix is propagated up to a matching distance and the asymptotic boundary conditions are

employed to extract the scattering matrix (S -matrix).

However, since all the K -labeled channels are coupled in the matrix, the computation becomes unaffordable for calculations with large J value. In this work, the coupled-states approximation including only the nearest neighbor Coriolis couplings (CSA-NNCC) approach is thus employed in the dynamical calculations. In this approximation approach, the propagation is

divided into K -labeled processes, in which only the nearest neighbor angular basis functions from $K-1$ to $K+1$ are included. This approach is better than the original coupled states approximation (CSA)^{8,9}, in which all Coriolis couplings are discarded. After each propagation, only the S -matrix elements of K -initial state in each K -labeled propagation are used to generate the state-to-state probability.

The transition probability from the initial $(v_1, j_1; v_2, j_2)$ to a final one $(v'_1, j'_1; v'_2, j'_2)$ CMSs is given below:

$$P_{v'_1 j'_1 v'_2 j'_2 \leftarrow v_1 j_1 v_2 j_2}^J(E_c) = \frac{(1 + \delta_{v_1 v_2} \delta_{j_1 j_2})(1 + \delta_{v'_1 v'_2} \delta_{j'_1 j'_2})}{2(2j_1 + 1)(2j_2 + 1)} \times \sum_{P_{\text{ex}} p \varepsilon} \sum_K \sum_{K'=K-1}^{K+1} \sum_{j_{12} j'_{12}} |S_{v'_j K', v_j K}^{\pm J p \varepsilon(K)}(E_c)|^2. \quad (14)$$

Finally, we note that the factor of 2 in the denominator of Supplementary Eq. (14) originates from the equal weight of $P_{\text{ex}} = \pm 1$ cases, which is different from the algorithm employed in the $\text{H}_2\text{-H}_2$ system¹⁰.

The validity of the CSA-NNCC approach for this system was re-examined. We calculated state-to-state probabilities for $J(P_{\text{ex}} p \varepsilon) = 10(+-)$ at the total energy of $E = 1.133$ eV, which makes the collision energy $E_c = 0.123$ eV for the initial CMS $(1, 1; 0, 1)$, by using the rigorous CC, original CSA^{8,9}, and the CSA-NNCC approach¹¹, respectively. Taking the rigorous CC probabilities as reference, the RMSE from a single initial CMS to all final CMSs is evaluated as,

$$\text{RMSE}_{v_j} = \sqrt{\sum_{v'_j} \frac{(P_{v'_j \leftarrow v_j}^{\text{approx}} - P_{v'_j \leftarrow v_j}^{\text{rigorous}})^2}{N^2 - 1}}, P_{v'_j \leftarrow v_j}^{\text{rigorous}} > 0.01, \quad (15)$$

where N is total number of final CMSs whose probabilities are larger than 0.01, and $P_{v'_j \leftarrow v_j}^{\text{rigorous}}$ and $P_{v'_j \leftarrow v_j}^{\text{approx}}$ are probabilities of rigorous and approximation methods, respectively. Supplementary

Figure 2 shows RMSEs of all 107 open initial CMSs at the selected energy and parity, and they are arranged by their j_1+j_2 values from high to low. The RMSEs calculated out by CSA-NNCC approach are systematically smaller than those by CSA, which are sometimes large especially when the initial CMSs have high j_1+j_2 values. Since the RMSEs by CSA-NNCC are less than 0.01 in almost all cases, this approach is validated and thus employed in this work.

Supplementary Notes 2

The quasi-classical trajectory (QCT) calculations for the HF($v_1=0, j_1=0$)+ HF($v_2=0, j_2=0$) collision were performed using the VENUS code. The trajectories were initiated with 10 Å separation between two HF molecules, and terminated when they reached a separation of 10 Å again. The propagation time step was selected to be 0.1 fs which conserved the energies better than 0.01 kcal mol⁻¹ for all trajectories. The trajectories were halted if the propagation time is exceptionally long (10.0 ps). The maximal impact parameter b_{max} was determined by the maximal total angular momentum J_{max} ,

$$b_{max}^2 \approx J_{max}(J_{max} + 1)/k^2 \quad (16)$$

in which $k^2 \hbar^2 = 2\mu E_c$. Here, b_{max} are chosen to be 7.143 Å. In total, 19 316 trajectories have been calculated.

Supplementary Figure 3 displays the lifetime distributions for trajectories with different numbers of inner turning points at $E_c=0.05$ kcal/mol. There are about 10% of trajectories undergo trapping with more than one inner turning point. These trajectories have significantly longer lifetimes, enabling energy randomization.

Supplementary References

1. Wu, Q., Zhang, D. H. & Zhang, J. Z. H. 6D quantum calculation of energy levels for HF stretching excited (HF)₂. *J. Chem. Phys.* **103**, 2548-2554 (1995).
2. Zhang, J. Z. H. *Theory and Application of Quantum Molecular Dynamics*. World Scientific (1999).
3. Zare, R. N. *Angular Momentum*. Wiley (1988).
4. Lin, S. Y. & Guo, H. Full-dimensional quantum wave packet study of rotationally inelastic transitions in H₂ + H₂ collision. *J. Chem. Phys.* **117**, 5183 (2002).
5. Zhang, D. H., Wu, Q., Zhang, J. Z. H., von Dirke, M. & Bačić, Z. Exact full-dimensional bound state calculations for (HF)₂, (DF)₂, and HFDF. *J. Chem. Phys.* **102**, 2315 (1995).
6. Johnson, B. R. The multichannel log-derivative method for scattering calculations. *J. Comput. Phys.* **13**, 445-449 (1973).
7. Manolopoulos, D. E. An improved log derivative method for inelastic scattering. *J. Chem. Phys.* **85**, 6425-6429 (1986).
8. McGuire, P. & Kouri, D. J. Quantum mechanical close coupling approach to molecular collisions. j₂-conserving coupled states approximation. *J. Chem. Phys.* **60**, 2488-2499 (1974).
9. Pack, R. T. Space-fixed vs body-fixed axes in atom-diatom molecule scattering. Sudden approximations. *J. Chem. Phys.* **60**, 633-639 (1974).
10. Quémener, G. & Balakrishnan, N. Quantum calculations of H₂-H₂ collisions: From ultracold to thermal energies. *J. Chem. Phys.* **130**, 114303 (2009).
11. Yang, D., Hu, X., Zhang, D. H. & Xie D. An improved coupled-states approximation including the nearest neighbor Coriolis couplings for diatom-diatom inelastic collision. *J. Chem. Phys.* **148**, 084101 (2018).